# Phase diagram of $Bi_2Sr_2CaCu_2O_{8+\delta}$ revisited

I.K. Drozdov [1], I. Pletikosić [1,2], C.-K. Kim[1], K. Fujita[1], G.D. Gu[1], J.C.Séamus Davis[1,3], P.D. Johnson[1], I. Božović [1] & T. Valla [1]

In cuprate superconductors, the doping of carriers into the parent Mott insulator induces superconductivity and various other phases whose characteristic temperatures are typically plotted versus the doping level $p$. In most materials, $p$ cannot be determined from the chemical composition, but it is derived from the superconducting transition temperature, $T_c$, using the assumption that the $T_c$ dependence on doping is universal. Here, we present angle-resolved photoemission studies of $Bi_2Sr_2CaCu_2O_{8+\delta}$, cleaved and annealed in vacuum or in ozone to reduce or increase the doping from the initial value corresponding to $T_c = 91$ K. We show that $p$ can be determined from the underlying Fermi surfaces and that in-situ annealing allows mapping of a wide doping regime, covering the superconducting dome and the non-superconducting phase on the overdoped side. Our results show a surprisingly smooth dependence of the inferred Fermi surface with doping. In the highly overdoped regime, the superconducting gap approaches the value of $2\Delta_0 = (4 \pm 1)k_B T_c$

[1] Condensed Matter Physics and Materials Science Department, Brookhaven National Lab, Upton, NY 11973, USA. [2] Department of Physics, Princeton University, Princeton, NJ 08544, USA. [3] Laboratory of Atomic and Solid State Physics, Department of Physics, Cornell University, Ithaca, NY 14853, USA. Correspondence and requests for materials should be addressed to T.V. (email: valla@bnl.gov)

Bi$_2$Sr$_2$CaCu$_2$O$_{8+\delta}$ (Bi2212) is a prototypical cuprate high-Tc superconductor (HTSC) and one of the most studied materials in condensed matter physics. Its phase diagram has been heavily studied by many different techniques. In particular, Bi2212 has been the perfect subject for studies by Angle Resolved Photoemission Spectroscopy (ARPES) and Spectroscopic Imaging - Scanning Tunneling Microscopy (SI-STM) due to its ease of cleaving. These techniques have contributed significantly to our current understanding of the cuprates by providing invaluable information on different phenomena and their development with doping, mostly in Bi2212. The d-wave symmetry of the superconducting (SC) gap[1,2], the normal state gap (pseudogap)[3–5], the quasiparticle (QP) self-energy[2,6,7] and, more recently, the transition in topology of the Fermi surface (FS) from open to closed[8,9] are a few notable examples of such contributions. However, Bi2212 can only be doped within a relatively limited range, especially on the overdoped side, where the superconducting transition temperature ($T_c$) cannot be reduced beyond ~50 K, leaving a large and important region of the phase diagram out of reach. Even within the covered region, the actual doping level $p$ is not independently determined, but it is usually calculated from the measured $T_c$ by assuming the putative parabolic $T_c - p$ dependence that is considered universal for all the cuprates[10]. Only in a very limited number of materials such as La$_{2-x}$Sr$_x$CuO$_4$ and La$_{2-x}$Ba$_x$CuO$_4$, can the doping be approximately determined from chemical composition as $p \approx x$. These two materials however, have very different $T_c - p$ dependences, illustrating the invalidity of the universal parabolic $T_c - p$ dome[11].

Here, we revisit the Bi2212 phase diagram by modifying the doping level of the as-grown crystal, by annealing in situ cleaved

samples either in vacuum, resulting in homogeneous under-doping, or in ozone, resulting in overdoping of the near-surface region. We were able to span a wide region of the phase diagram, ranging from strongly underdoped, with the $T_c$ reduced down to 30 K (UD30), to strongly overdoped, where the superconductivity was completely suppressed (OD0). In addition, we were able to infer the doping level directly from ARPES, by measuring the volume of the underlying FS. In that way, we follow the development of spectral features with doping with unprecedented clarity and detail and construct the phase diagram of Bi2212.

## Results

**Fermi surface.** Figure 1 shows the photoemission intensity integrated within ±2.5 meV around the Fermi level for the as-grown sample and for several different doping levels, induced by annealing in vacuum, or ozone. Due to the presence of the spectral gap, only the near-nodal segments are visible, while the intensity is strongly reduced or absent in most of the Brillouin zone (BZ), except in the highly overdoped samples with reduced gaps. As we move down in energy, more and more of the missing intensity is recovered, and at the energy of the maximal spectral gap ($\Delta_0$), the full, closed contour of the underlying FS is recovered. Although the Luttinger theorem requires that, to determine the carrier density, one measures the closed contours only for QPs at $E = 0$ (Fermi level), this is rendered impossible by the existence of the spectral gap. Therefore, we hypothesize that the contour of minimum gap energy may be used for the same purpose. This hypothesis assumes particle-hole symmetry of the gapped state about the chemical potential. In that case, connecting the ($k_x$, $k_y$) points where the intensity first appears, yields the curves of minimal

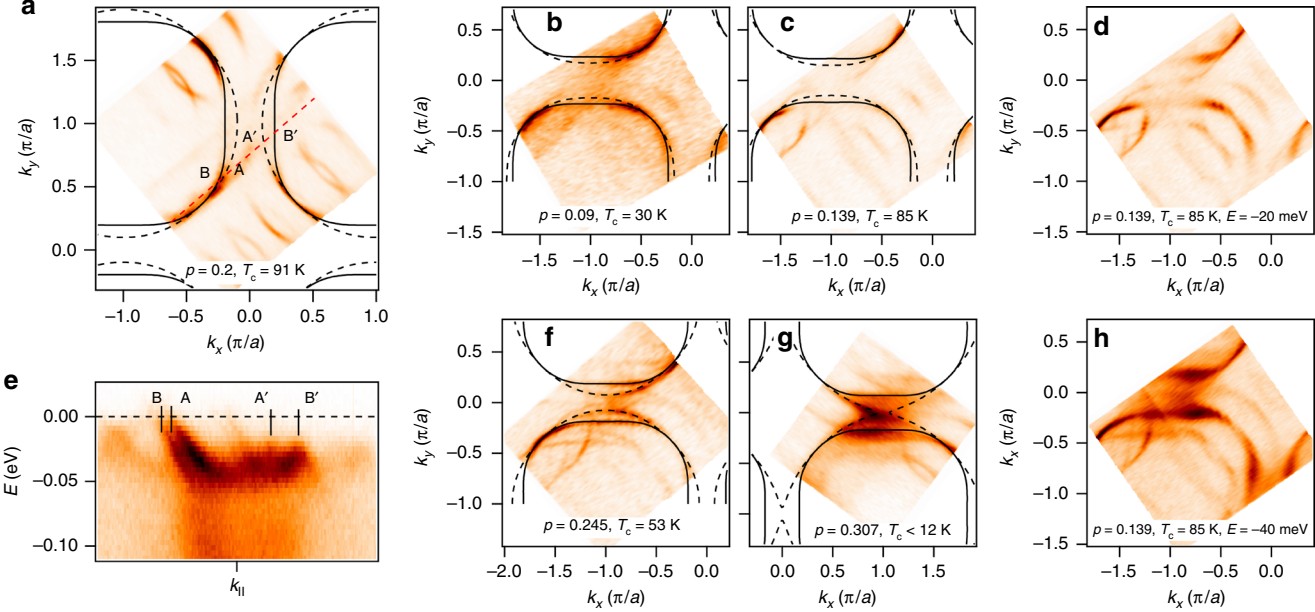

**Fig. 1** Development of the electronic structure of Bi$_2$Sr$_2$CaCu$_2$O$_{8+\delta}$ at the Fermi level with doping. **a** Fermi surface of an as grown sample. **b**, **c** Underdoped samples, obtained by annealing of the as grown samples in vacuum. **f**, **g** Overdoped samples, obtained by annealing of as grown samples in O$_3$. Spectral intensity, represented by the false color contours, is integrated within ±2.5 meV around the Fermi level. Solid (dashed) lines represent the bonding (antibonding) states obtained from the tight-binding approximation that best fits the experimental data. The fitting involved the lines connecting the experimental minimal gap loci in all cases where the underlaying Fermi surface was gapped, as indicated in **e**. **e** Dispersion of states along the red dashed line in **a**. Fermi momenta of bonding (B, B′) and antibonding (A, A′) are marked in both **a** and **e**. Gapped Fermi momenta (A′, B′) were approximated by the points where the dispersion acquire maximum. **d** Intensity contour for sample UD85 (from **c**) at $E = -20$ meV and **h** $E = -40$ meV. The area enclosed by the TB lines that best represent the experimental data is calculated and used for determination of the doping parameter $p_A$. The superconducting transition temperature, $T_c$, is determined from magnetic susceptibility measurements (as grown and underdoped samples) and from ARPES data, as explained in the text. All the maps were recorded at 12–20 K

gap loci, which we refer to throughout as the underlying FSs. Mapping at higher temperatures, above $T_c$ and above the pseudogap temperature, $T^*$, that was performed at several doping levels (see Methods section), shows exactly the same Fermi contours, supporting our approach. We do not see any change in shape between the low-temperature and high-temperature Fermi contours that were previously reported in ref. [12]. Figure 1 also shows tight binding (TB) contours, with solid (dashed) lines representing the bonding (antibonding) states[13–15]. The TB parameters for several doping levels are given in Table 1. The number of carriers forming the underlying FS is obtained from the Luttinger count of the area enclosed by the Fermi contour, $p_L = 2A_{FS}$. To express this in terms of doping that usually serves as the abscissa in phase diagrams of the cuprates, we calculate the number of additional holes doped into the parent Mott insulator as $p_A = p_L - 1 = 2A_{FS} - 1$. Here, both the bonding and the antibonding states are counted, $A_{FS} = (A_B + A_A)/2$, originating from two Cu–O planes per unit cell, and the area of the BZ is set to one. We now have an independent doping parameter for Bi2212 and do not have to rely on the universal dome when plotting the phase diagram.

We note that $p_A$ differs from the concentration of mobile carriers contributing to transport, $p_{tr}$, a quantity that has been difficult to measure directly and accurately. However, recent models and high field studies have indicated that the mobile hole density switches from $p_{tr} \sim p_A$ in the underdoped samples to $p_{tr} \sim p_L = p_A + 1$ near optimal doping[16–18].

In this study, the underlying FS shows a smooth development with doping, with no evidence for the transition from a small Fermi pocket to big FS. Due to the structural supermodulation, multiple diffracted replicas of the intrinsic band structure are observed in addition to the "shadow" FS, shifted by $(\pi/a, \pi/a)$[13,14]. The intrinsic underlying FS keeps the same topology across most of the doping regime: the two big FSs, corresponding to the bonding and antibonding states, resolved in all but the least doped (UD30) sample, remain hole-like. Only at the highest doping levels, where the superconductivity gets suppressed beyond our limits of detection, the antibonding FS undergoes the Lifshitz transition and becomes an electron pocket centered at (0, 0). No conventional reconstruction of the underlying FS is visible in our data that we can assign to the charge or spin orders observed in some other cuprates[18–21]. On the contrary, the observed smooth development with doping would imply that the underlying electronic structure of Bi2212 could be approximated by the same TB bands filled to different levels (Table 1). Also, the effects of charge ordering observed in the single-layer compound (Bi, Pb)$_2$(Sr,La)$_2$CuO$_{6+\delta}$ by Comin et al.[22], but not by Kondo et al.[23], do not show up in any of our samples. We also do not observe changes in $k$-space coherence of the low-energy QP excitations when they cross the antiferromagnetic zone boundary[9]. In all samples studied here in the SC phase, QPs retain coherence on both sides of this boundary.

Our study, however, does confirm that the actual FS of Bi2212 is strongly affected by the pseudogap, i.e., it is partially gapped in the normal state in a wide portion of the phase diagram[2–4,8]. Our findings are consistent with the general notion that the electronic states cross the Fermi level only in the nodal region, forming the so-called Fermi arcs that grow in length with increasing doping and eventually enclose the whole underlying FS. Whether these arcs arise due to the gap closing, or its filling is still debated[24–26], but the general picture is in line with modern $k$-resolved calculations based on Mott physics that reproduce the nodal-antinodal dichotomy observed in

**Table 1 Tight binding parameters for Bi$_2$Sr$_2$CaCu$_2$O$_{8+\delta}$**

| Sample | $\mu$ (meV) | $t$ (meV) | $t'$ (meV) | $t''$ (meV) | $t_\perp$ (meV) |
|--------|------|------|------|------|------|
| UD30 | 340 | 390 | 120 | 45 | 108 |
| UD78 | 330 | 350 | 100 | 45 | 95 |
| OD91 | 405 | 360 | 108 | 36 | 108 |
| OD67 | 430 | 360 | 108 | 36 | 108 |
| OD35 | 445 | 360 | 108 | 36 | 108 |
| OD0 | 467 | 360 | 108 | 36 | 108 |

experiments[27]. We note that the Fermi arcs cannot be distinguished from the small Fermi pockets in which only one side is observable, while the other is invisible due to the zero spectral weight[8,9,28]. For similar reasons, the observed maps cannot exclude the existence of a pair density wave state, as the relevant portions of reconstructed FS also carry a vanishing spectral weight in that scenario[29]. To differentiate between these pictures, further theoretical and experimental studies will be required.

**Spectral gap**. In the following, we extract the spectral gap and $T_c$ of each modified sample in order to reconstruct the appropriate phase diagram for Bi2212. Vacuum annealing results in a seemingly homogeneous doping profile with the surface $T_c$ measured by ARPES showing good agreement with the bulk susceptibility measurements. Thus, on the underdoped side, we usually have two independent measurements of $T_c$. However, annealing in ozone results in increased doping only in the near-surface region. Therefore, the only measure of $T_c$ in the overdoped regime was spectroscopic: the temperature induced changes in the QP peak intensity, as well as the leading edge position clearly indicate $T_c$[2,26].

Figure 2 shows the total spectral gap from the antinodal region ($\Delta_0$) and its temperature dependence for several selected samples with different doping levels. At the antinodal point the $d$-wave SC gap has its maximal amplitude. The normal state gap (pseudogap) is also maximal there[2–4,8,30]. The energy distribution curves from that point, shown in Fig. 2f, clearly indicate that the QP peak is shifting closer to the Fermi level as the doping increases, reflecting the reduction of the spectral gap. Also shown is the typical temperature dependence of the spectra for an underdoped sample (UD78) and an overdoped sample (OD62). For the underdoped sample, the gap does not close at $T_c$ determined from susceptibility measurements, but at some higher temperature $T^*$. However, the leading edge gap and intensity of the QP peak show a prominent change around $T_c$ and the later could be identified as being near the inflection point of these temperature dependencies (Fig. 2i, j)[26].

**Phase diagram**. The antinodal gap magnitude, $\Delta_0$ is determined from the symmetrized spectra as the position of the QP peak and plotted in Fig. 3 for all the doping levels covered in this study. The $T_c$ is also plotted. We first note that most of our data agree well with previous studies; gaps and transition temperatures from the present study follow the same trends with very little discrepancies[2,5,31–36]. The only adjustment needed for the perfect overlap with the previous studies was the shift in the doping level $p_A$. The SC dome for Bi2212 is now centered at $p_A = 0.18$. Its width remained approximately the same, in agreement with another study that reported the shift[37]. The data from the literature are placed in accordance to the new doping scale.

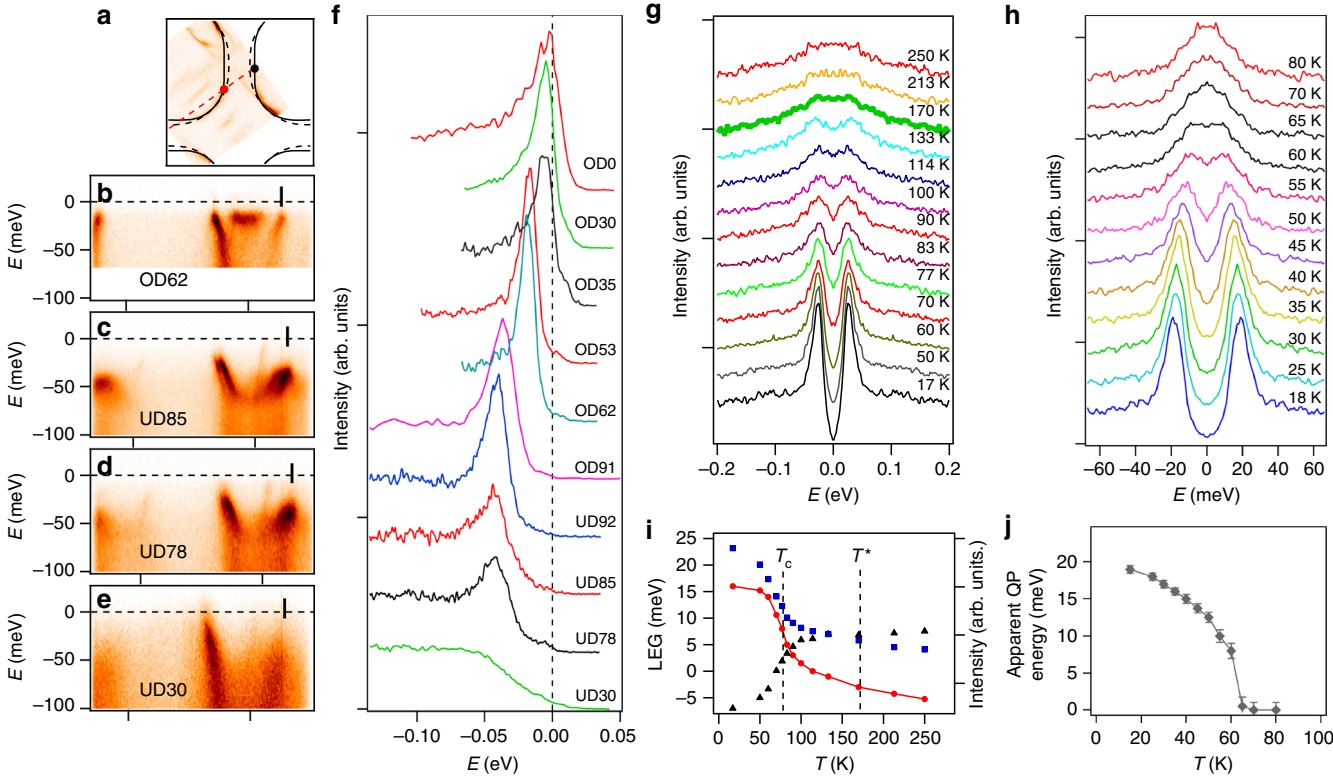

**Fig. 2** Spectral gap from the anti-nodal region of Bi$_2$Sr$_2$CaCu$_2$O$_{8+\delta}$. **a** Fermi surface of the UD85 sample, with the dashed red line indicating the momentum line that is probed in **b**–**e** for samples with different oxygen content. **b**–**e** ARPES intensity as a function of binding energy and momentum along the dashed line in **a**, taken at $T = 12$ K, for several different doping levels, as indicated. **f** The EDC curves taken at the Fermi wave-vector of the bonding state from the anti-nodal region of the Fermi surface, as indicated by bars in **b**–**e**. **g** Temperature dependence of ARPES spectra for the underdoped sample, UD78, ($T_c =$ 78 K) and **h** for the overdoped sample, OD62 ($T_c = 62$ K). The spectra in **g** and **h** are taken at $k_F$ marked with red and black dots in **a**, respectively. Spectra in **g**, **h** are symmetrized relative to the Fermi level. **i**, **j** Temperature dependence of several spectral parameters for the two samples shown in **g**, **h**: intensity at the Fermi level (black triangles) and at the peak energy (blue squares), Leading edge gap (LEG) of non-symmetrized spectra (red circles) and apparent quasiparticle peak position (gray diamonds). The error bars in **j** correspond to the fitting uncertainties in the apparent quasiparticle peak positions from **h**

## Discussion

The important new ingredient of the re-drawn phase diagram (Fig. 3) is the completely new doping regime that is covered by this study (represented with the shaded area in Fig. 3a): the highly overdoped regime, far beyond what was previously explored in Bi2212, has now been mapped, including the critical point at which superconductivity vanishes. As shown in Fig. 3b, it is only in this regime ($p_A > 0.25$) that $2\Delta_0/(k_B T_c)$ actually acquires low values consistent with those inferred by extrapolation from the less overdoped regime in previous studies[38].

With further increase in doping, both the gap and $T_c$ vanish around $p_A = 0.29$ and Bi2212 is no longer a superconductor. It is intriguing that this coincides with the change in FS topology: the antibonding FS undergoes a conventional Lifshitz transition, turning from a hole-like into an electron-like, as its saddle points at ($\pm\pi/a$, 0 and (0, $\pm\pi/a$) move above the Fermi level. The coincidence of these two transitions hints at the importance of the FS topology for superconductivity in the cuprates, but its exact role remains unclear. In the single layer counterpart, (Bi,Pb)$_2$(Sr,La)$_2$CuO$_{6+\delta}$, the Lifshitz transition is also coincidental with vanishing superconductivity[23]. This material, moreover, shows much more drastic departure from the universal SC dome, with the maximal $T_c$ occurring near $p_A = 0.3$.

In Bi2212, the Lifshitz transition affects only one FS and not the other. This would suggest that the changes in the antibonding state are for some reason more important for superconductivity than those in the bonding state. At this moment, it is not clear why this should be the case. One suggestion could be that the straight segments of the bonding states FS are susceptible to nesting and are involved in charge, spin or pair density waves, instead of superconductivity. However, in this highly overdoped regime, the lack of a gap and other effects that these phenomena should induce into the spectral response certainly argues against this scenario. Still, it seems that the curvature (or a lack of it) of the FS in the antinodal region plays a significant role in cuprate superconductivity and that the straight antinodal segments of the FS might be ineffective in forming the efficient singlet pairs even when not contributing to density waves. This might be because the group velocity of these straight segments lacks the component that would connect them from $k_F$ to $-k_F$ into singlet pairs. Further studies on other multi-layered cuprate superconductors will be necessary to resolve these questions.

## Methods

**Sample preparation.** The experiments within this study were done in a new experimental facility that integrates oxide-MBE with ARPES and STM capabilities

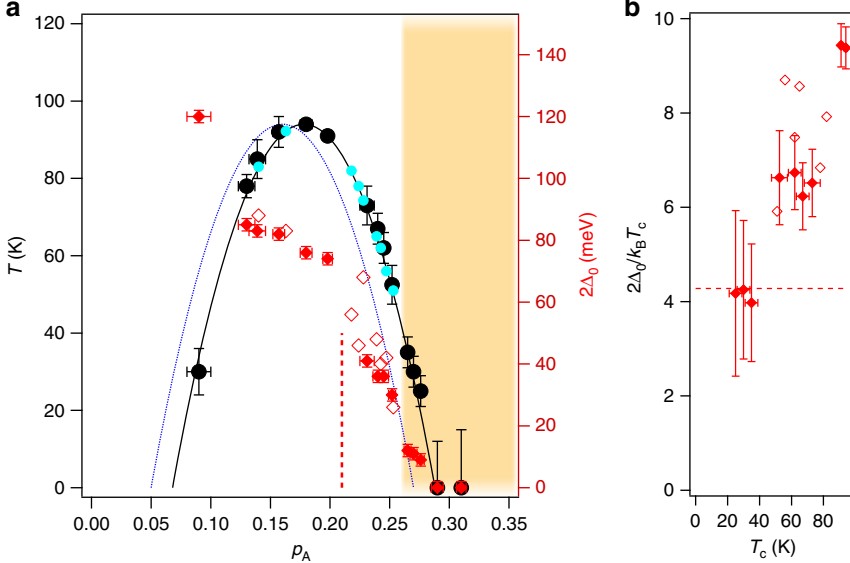

**Fig. 3** Phase diagram of $Bi_2Sr_2CaCu_2O_{8+\delta}$. **a** Experimentally obtained superconducting transition temperature, $T_c$, (black circles) and the antinodal gap, $2\Delta_0$ (solid red diamonds) are plotted versus experimentally determined doping parameter, $p_A$. Also shown are gap values (open red diamonds) and $T_c$ (turquoise circles) for different doping levels from several previously published studies[5,31,32,34]. These points are placed at the corresponding $p_A$ in accordance with our new $T_c(p_A)$ dependence. The universal superconducting dome is shown as blue dashed line, while the corrected dome is represented by black solid line. The red dashed line indicates the transition observed in SI-STM studies[9]. The shaded area marks the region that has not been studied before. Characteristic temperatures are displayed against the left axis, while the right axis is for the spectral gap. **b** Ratio of the antinodal gap and $T_c$ for the points shown in **a** from the overdoped side. The red dashed line corresponds to the BCS value for $d$-wave gap. We have approximated the uncertainty in doping, $\Delta p_A$, (horizontal error bars in **a**) to be proportional to the width of the Fermi surface: $\Delta p_A/p_A \sim 2\Delta k_F/k_F$. The uncertainty in $T_c$ is given by the width of superconducting transition in susceptibility measurements (underdoped samples), or by the temperature step size in $T$-dependent ARPES measurements that identify $T_c$ (overdoped samples). The uncertainty in gap magnitude (red vertical error bars in **a**) corresponds to the standard deviation of the quasiparticle peak position determined from fitting. It serves to determine the propagated uncertainty (vertical error bars) in **b**

within the common vacuum system[39]. The starting samples were slightly over-doped ($T_c = 91$ K) single-crystals of Bi2212, synthesized by the traveling floating zone method. They were clamped to the sample holder and cleaved with Kapton tape in the ARPES preparation chamber (base pressure of $3 \times 10^{-8}$ Pa). Thus, the silver-epoxy glue, commonly used for mounting samples and associated processing at elevated temperatures, has been completely eliminated, resulting in perfectly flat cleaved surfaces and unaltered doping level.

The cleaved samples were then annealed in situ in the ARPES preparation chamber to different temperatures ranging from 150 to 700 °C for several hours, resulting in the loss of oxygen and underdoping. For the overdoping, the cleaved as-grown samples were transfered to the MBE chamber (base pressure of $8 \times 10^{-8}$ Pa) where they were annealed in $3 \times 10^{-3}$ Pa of cryogenically distilled $O_3$ at 350–480 °C for $\approx 1$ h. After the annealing, films were cooled to room temperature in the ozone atmosphere and transfered to the ARPES chamber (base pressure of $8 \times 10^{-9}$ Pa). Vacuum annealing results in generally homogeneous doping profile where the surface $T_c$ measured by ARPES shows no variation with repeated re-cleaving of the annealed crystals and is in a good agreement with the bulk susceptibility measurements. Annealing of as grown crystals in $O_3$ results in increased doping in the near-surface region, as evidenced by the increased hole FS, reduced spectral gap and its closing temperature. The most of the crystals volume remained near the optimal doping upon ozone annealing. The thickness of the overdoped surface layer was in the sub-micron range, as only the thinnest, semi-transparent re-cleaved flakes showed the significant reduction in $T_c$ in susceptibility measurements.

**ARPES.** The ARPES experiments were carried out on a Scienta SES-R4000 electron spectrometer with the monochromatized HeI (21.22 eV) radiation (VUV-5k). The total instrumental energy resolution was ~5 meV. Angular resolution was better than ~0.15° and 0.4° along and perpendicular to the slit of the analyzer, respectively.

The ARPES estimate of $T_c$ of the overdoped surfaces was within ±4 K, except for the two samples falling outside of the SC dome, for which the estimate was limited by the base temperature that could be reached with our cryostat (12 and 15 K, for these two cases).

Maps of the underlying FSs in the SC state (gapped state) agree perfectly well with fully enclosed FS contours obtained in the state without any spectral gaps at $T > T^* > T_c$ as was checked for several samples, including the OD91, UD87 (Fig. 4) and several overdoped samples. Positions of minimal gap loci in the SC state overlap with the positions of Fermi momenta in the normal (gapless) state within the experimental resolution.

**TB parameters.** The bare in-plane band structure of Bi2212 used to fit the experimental FS contours is approximated by the tight-binding formula:

$$E_{A,B}(k) = \mu - 2t\left(\cos k_x + \cos k_y\right) + 4t'\cos k_x\cos k_y - 2t''\left(\cos 2k_x + \cos 2k_y\right) \pm t_\perp\left(\cos k_x - \cos k_y\right)^2/4, \quad (1)$$

where the index A (B) is for anti-bonding (bonding) state and $\mu$ is chemical potential. The hopping parameters that best describe the FSs of selected measured samples are given in Table 1.

The TB contours that agree with the experimental contours the best were chosen by eye. By changing them to the point where discrepancies would become clearly visible, we can estimate that the uncertainty in doping, $\Delta p_A$, of this method is very close to that estimated from the experimental momentum width of the FS, $\Delta p_A/p_A \sim 2\Delta k_F/k_F$.

The electronic structure of Bi2212 could be approximated by nearly the same TB parameters in the whole doping range covered in this study, indicating that our data show no evidence for dramatic transitions involving charge/spin ordered states with substantial spectral weight redistribution as doping is swept from strongly underdoped ($p = 0.09$) to strongly overdoped, non-superconducting ($p > 0.29$) regime. This observation suggests that in the studied doping range, the underlying electronic structure of Bi2212 can, in essence, be described as the progressive filling of the same bilayer split bands, with spectral gaps and pseudogaps as additional necessary ingredients in SC samples, in line with the recent proposal by Pelc et al.[40].

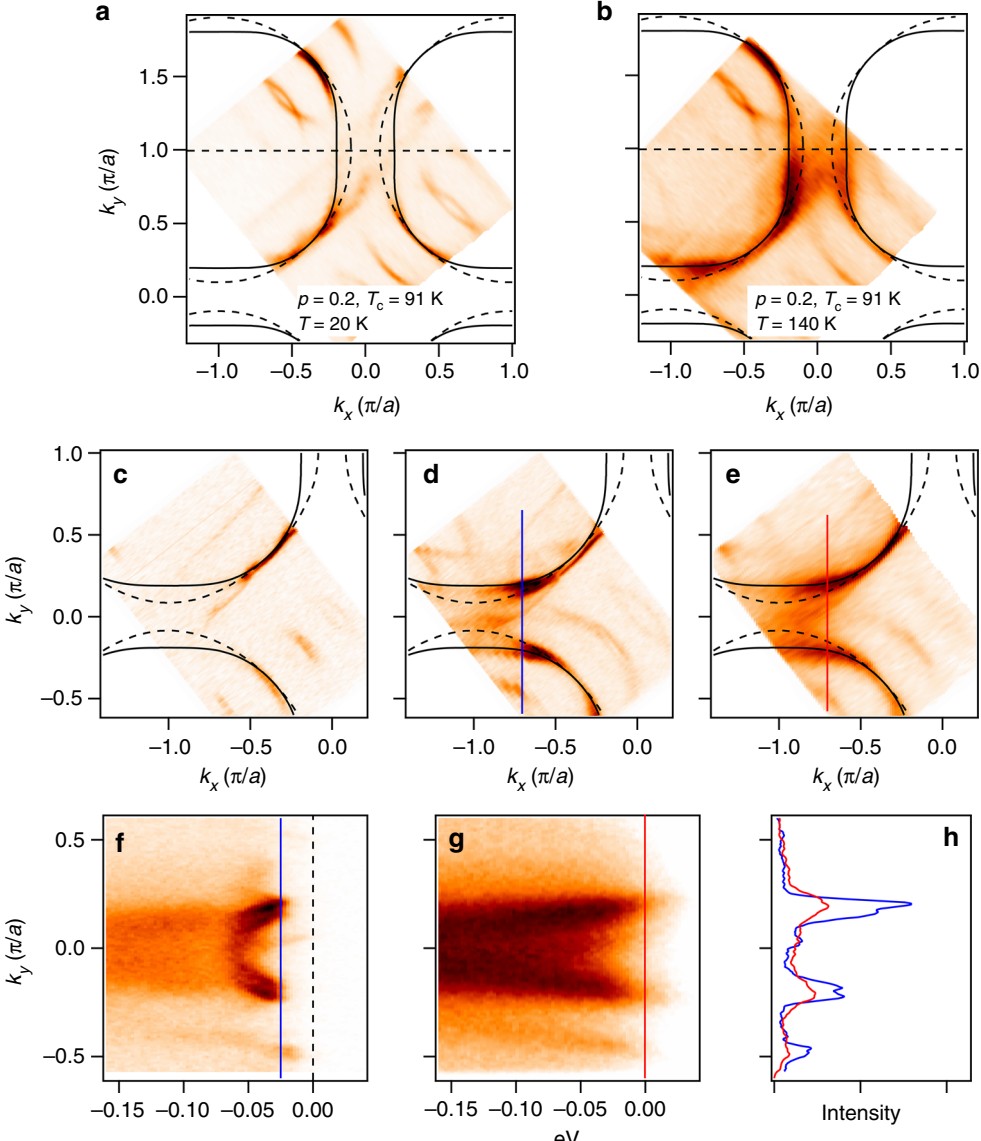

**Fig. 4** Comparison of the underlying gapped FS with the gapless FS. **a** FS contour in the superconducting state and **b** fully enclosed FS contour obtained in the normal state, at $T = 140$ K ($>T^*$) for the as grown sample, OD91. The TB parameters are identical in both panels. **c** Constant energy contour at $E = 0$ and **d** at $E = -25$ meV in the superconducting state ($T = 20$ K) of the underdoped sample, UD87. **e** The normal state contour at $E = 0$ of the same sample ($T = 125$ K). The TB contours are identical in **c–e**. **f**, **g** show the photoemission intensities along the momentum lines indicated in **d**, **e**, $k_x = -0.7\ \pi a^{-1}$ in the superconducting and normal states, respectively. **h** Momentum distribution curves at $E = -25$ meV and at $E = 0$, with the maxima corresponding to the gapped and gapless Fermi momenta from **f** and **g**, respectively

## Data availability

The data that support the findings of this study are available from the corresponding author upon reasonable request. The source data underlying Figs. 2j and 3a, b are provided as a Source Data file.

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

## Acknowledgements

We thank J. Tranquada, R. Konik, and J. Misewich for discussions. This work was supported by the US Department of Energy, Office of Basic Energy Sciences, contract no. DE-AC02-98CH10886. I.P. is supported by ARO MURI program, grant W911NF-12-1-0461. I.K.D. acknowledges the generous financial support of the BNL Gertrude and Maurice Goldhaber Distinguished Fellowship.

## Author contributions

I.K.D. and T.V. designed and directed the study. G.D.G. grew the bulk crystals and performed magnetization measurements. I.K.D. performed the sample preparation in ozone. I.P. and T.V. performed the ARPES experiments. T.V. analyzed and interpreted data and wrote the manuscript. I.K.D., I.P., C.-K.K., K.F., J.C.S.D., P.D.J., I.B. and T.V. made contributions to development of the OASIS facility used herein and commented on the manuscript.

## Additional information

**Competing interests:** The authors declare no competing interests.

