## [Peer Review File · Nature Communications]

Reviewers' comments:

Reviewer #1 (Remarks to the Author):

The paper presents the most careful ARPES study of the bi-layer Bi-based high temperature superconductor in a widest doping range. It contains several key results, which contribute significantly to our current understanding of the cuprates, but not all of them are really stressed. It shows that (1) the FS contours do not change with temperature being the same in the normal state, below T^* , and below T_c , and the tight binding parameters are derived for many doping levels; (2) the bi-layer splitting is present in the whole doping range; (3) the topological Lifshitz transition for the anti-bonding band coincides with the overdoped edge of the superconducting dome.

All these points were subjects of long standing controversies which are not mentioned here. On the other hand, it would require including many additional references and discussions that would greatly complicate the article.

These results have been obtained due to in-situ annealing procedure in a new experimental facility that integrates oxide-MBE with ARPES and STM capabilities within the common vacuum system that is unique at the moment.

The results presented in the paper are of immediate interest to many people in the field of high temperature superconductivity. The paper is clearly written and the reporting of data and methodology is sufficiently detailed and transparent. I recommend its publication in Nature Communications.

Nevertheless, there are some points that, if addressed by Authors, should improve the paper.

The uncertainty in determination of the FS position and the doping level, respectively, should be much higher in the vicinity to topological transition due to low Fermi velocity, but the horizontal error bars in Fig.3A do not reflect it.

It would be useful to give the uncertainty for the TB parameters in Table 1. In particular, to address the question whether one can conclude about the rigid shift with doping within the experimental uncertainty.

Reviewer #2 (Remarks to the Author):

This experimental work from Drozdov and co-workers presents an angle-resolved photoemission spectroscopy (ARPES) study of Bi2212 single crystals, asserting this to be the correct method for determination of doping level. Nearly-optimally-doped Bi2212 single crystals were annealed in-situ (in vacuum or ozone) to adjust the doping level, the value of which was then extracted from the ARPES-measured Fermi surface area via application of the Luttinger theorem. The Authors were able to redefine the superconducting dome in the phase diagram of Bi2212, present the first ARPES characterization of the very overdoped (OD) region of the superconducting dome and gain new insights regarding the importance of the Fermi surface topology for superconductivity in cuprates.

The Authors address the need for a tool to determine the doping level of Bi2212 crystals (and potentially for all the cuprates), instead of relying on the commonly employed (and not necessarily valid) universal relation between T_c and doping level. I agree that this is an interesting result for the cuprate community. Therefore, I consider this work suitable for the publication in Nature Communications after addressing/clarifying these following points.

1. The Authors claim that the “minimal gap loci” curves map the Fermi surface effectively, facilitating application of the Luttinger theorem. In addition, the Authors remark that this method is valid only if particle-hole symmetry is verified. However, there are several experiments which suggest that the pseudogap is not particle-hole symmetric (e.g. Ref. [12], discussed in the following). How would the particle-hole asymmetry of the pseudogap influence the extracted doping level (relative error)? How might different origins of the pseudogap impact the applicability of the Luttinger theorem?

Along the same line, the assumption of particle-hole symmetry at the antinode is not well-founded unless the pseudogap is a precursor to the superconducting state (which does not seem to be the case, as pre-formed Cooper pairs are detected for temperatures well below T^* , c.f. Ref. [26] or Reber et al., Nat Phys 8, 606 (2012)). Therefore, the assumption of particle-hole symmetry may lead to the extraction of an erroneous gap amplitude at the antinode. In addition, at the antinode both the superconducting gap and pseudogap are likely superimposed: consequently, I think that the plot of $2\Delta_0$ as the function of the doping in Fig.3a and Fig.3b is not relevant at this stage. There is already confusion within the community regarding determination of the antinodal gap value and its origin. I advise the Authors to reword/change this part of the manuscript.

However, I find it very interesting that for highly OD samples, i.e. where the pseudogap is absent and the $2\Delta_0$ value can be associated directly with the superconducting gap, the value of $2\Delta_0/kBT_c$ is close to what is expected for a d-wave BCS.

2. The Authors claim, in just one sentence, that they “do not see any change in shape between the low- and high-temperature Fermi contours that were reported in Ref. [12]”. This is a very important point, essential to validate the methodology proposed in the manuscript and should be discussed in more depth. I recommend the Authors add additional experimental evidence supporting their claim (for instance EDCs at various momenta and temperatures as shown in Ref. [12], as well as the suggestion below in minor point i.).

3. The Authors do not provide error bars for the extracted doping levels (p). Error bars and how they have been determined should be added to the main text. In addition, how is the minimum gap loci determined (via EDCs or MDCs)? The Authors used a simplified tight-binding model from Ref. [13] but they neglect the (weak) k_z dependence: how could k_z affect the fitting and the relative error of the extracted doping level? How was the tight-binding fitted to the experimental data (by eye, maximum of the EDCs, etc...)? How did they distinguish bonding and antibonding states in very UD samples? Was the fitting procedure robust?

4. The authors can determine the T_c for OD samples annealed in ozone only through spectroscopic measurements. This is done by tracking the evolution of the low-energy one-electron removal spectral function as shown in Ref. [26]. However, Ref. [26] is not cited correctly. Kondo et al. (Ref. [26]) have shown that the superconducting gap is filled rather than closed due to the loss of coherence of the macroscopic superconducting condensate, for both UD and OD samples. Therefore, the quasiparticle peak position is not really moving. On the contrary, quasiparticle spectral weight moves from the quasiparticle peaks into the gap. This is clear looking at Fig. 2G. The authors should reword this section.

Ultimately, I agree that the determination of T_c via the leading edge position or the intensity at the Fermi level is sufficiently accurate for the purpose of this manuscript.

Additional (Minor) Points:

i. The Authors discuss in the main text and in Methods that the mapping for $T > T^*$ shows exactly the same Fermi contours observed for $T < T_c$. They show in Fig. S2 a comparison of two FS for $T = 20$ K and

$T=140$ K, for Bi2212 OP91. Given the fact that this observation is in stark contrast with what was reported in Ref. [12], I would recommend the Authors to show the same comparison for the other OD samples they measured, as well as UD85.

ii. References [14, 15] are placed during the discussion of the Luttinger count. These references are more appropriate for discussing the tight-binding model or the observed diffracted replicas and shadow bands.

iii. Dorion-Leyraud and co-workers (Nat. Comm. 8:2044) have shown that the pseudogap appears only for dopings $p < p_{\text{Lifshitz}}$. In the present case, experimental data suggest the pseudogap in Bi2212 disappears for OD60/OD80 (corroborated also by SI-STM studies, Ref. [9], dashed red line in Fig. 3A) while the Lifshitz transition occurs at the end of the superconducting dome. I feel that this point should be mentioned and discussed in more depth. The first ARPES study of the very OD part of the phase diagram and the observation of the critical point associated with the end of the superconducting dome are definitely interesting. Differences between the critical points associated with the end of the superconducting dome and the pseudogap should be discussed.

iv. Comin et al. [22] did not report any observable reconstruction of the Fermi surface but they suggested that the CDW order originates from the nesting vector connecting the end of the Fermi arcs.

v. The Authors discuss the presence of the pseudogap at the antinode and list several possible origins (CDW, SDW, PDW, YRZ model) even if direct evidence of Fermi surface reconstruction is not observed (possibly due to the vanishing of the spectral weight). Additional scenarios where the pseudogap is induced by strong-electronic correlations, i.e. the localization/delocalization of charges associated with the proximate Mott insulating phase, should be discussed briefly.

vi. As discussed previously, the superconducting gap for both OD and UD samples does not close at T_c (Ref. [26], Reber et al., Nat Phys 8, 606 (2012) or Boschini et al. Nat. Mat. 17, 416 (2018)). The superconducting gap closes at a higher temperature T_p different than T^* ($T_c < T_p < T^*$).

vii. In the caption of Fig. 2: spectral (G) and (H) are symmetrized relative... (missing (H)).

Reviewers' comments:

Reviewer #1 (Remarks to the Author):

The paper presents the most careful ARPES study of the bi-layer Bi-based high temperature superconductor in a widest doping range. It contains several key results, which contribute significantly to our current understanding of the cuprates, but not all of them are really stressed. It shows that

(1) the FS contours do not change with temperature being the same in the normal state, below T^* , and below T_c , and the tight binding parameters are derived for many doping levels;

(2) the bi-layer splitting is present in the whole doping range;

(3) the topological Lifshitz transition for the anti-bonding band coincides with the overdoped edge of the superconducting dome.

All these points were subjects of long standing controversies which are not mentioned here. On the other hand, it would require including many additional references and discussions that would greatly complicate the article.

We thank to the referee on these kind remarks. He/She is correct when suggesting that our results imply more than what was actually discussed in relation to some long-standing controversies of the cuprate physics. Our initial approach was to reduce the discussion to a minimum, but now we feel encouraged to expand on some of the most important results. For example, we now discuss the correlation between the Lifshitz transition and disappearance of superconductivity in more detail. We have also added a figure for a moderately underdoped sample to additionally back our conclusion that the gapped low- T contours are equivalent to the gapless contours at higher temperatures, $T > T^$. We now also discuss the observation that the electronic structure could be approximated by the same TB parameters virtually in the whole doping range covered in this study, indicating that our data show no evidence for dramatic transitions involving any charge/spin ordered states with substantial spectral weight redistribution as doping is swept from strongly underdoped ($p=0.09$) to strongly overdoped, non-superconducting ($p>0.3$) regime. This observation suggests that in the studied doping range, the underlying electronic structure of Bi2212 can, in essence, be described as the progressive filling of the same bilayer split bands, with spectral gaps and pseudogaps as additional necessary ingredients in superconducting samples.*

These results have been obtained due to in-situ annealing procedure in a new experimental facility that integrates oxide-MBE with ARPES and STM capabilities within the common vacuum system that is unique at the moment.

The results presented in the paper are of immediate interest to many people in the field of high temperature superconductivity. The paper is clearly written and the

reporting of data and methodology is sufficiently detailed and transparent. I recommend its publication in Nature Communications.

Nevertheless, there are some points that, if addressed by Authors, should improve the paper.

The uncertainty in determination of the FS position and the doping level, respectively, should be much higher in the vicinity to topological transition due to low Fermi velocity, but the horizontal error bars in Fig.3A do not reflect it.

The referee is correct in realizing that the Fermi surface volume displays the fastest change around the doping corresponding to the Lifshitz transition of the anti-bonding band. This, and decreasing Fermi velocity would naively imply a broader Fermi surface, resulting in increased uncertainty. In addition, a finite k_z hopping, that is expected to be maximal in the antinodal region, could also broaden the states and increase the uncertainty in determining the proper contour. However, in reality, the Fermi surface as seen in ARPES becomes sharper with increasing doping, due to the lower scattering rate and reduced coupling to other excitations in the system. We have approximated the uncertainty in p to be proportional to the width of the Fermi surface: $dp/p \sim 2dk_F/k_F$. There are other factors that may make the absolute determination of p less or more precise, but were not taken into account for the uncertainty calculations. For example, the band replicas, originating from supermodulation allow oversampling that increases precision. A small angular distortion near the edges of our maps might introduce a systematic error, but we found it to be insignificant in the tests where the same sample was mapped with several different azimuthal orientations.

It would be useful to give the uncertainty for the TB parameters in Table 1. In particular, to address the question whether one can conclude about the rigid shift with doping withing the experimental uncertainty.

By analyzing the shape of the low-energy constant energy contours, we have realized that, except for the most heavily underdoped samples, most of the TB parameters can be fixed, and the chemical potential is the only one that varies. The TB contours that agree with the experimental contours the best were chosen by eye. By changing them to the point where discrepancies would become clearly visible, we can estimate that the uncertainty of this method is very close to that estimated from the experimental momentum width of the Fermi surface, $dp/p \sim 2dk_F/k_F$ as already discussed.

Reviewer #2 (Remarks to the Author):

This experimental work from Drozdov and co-workers presents an angle-resolved

photoemission spectroscopy (ARPES) study of Bi2212 single crystals, asserting this to be the correct method for determination of doping level. Nearly-optimally-doped Bi2212 single crystals were annealed in-situ (in vacuum or ozone) to adjust the doping level, the value of which was then extracted from the ARPES-measured Fermi surface area via application of the Luttinger theorem. The Authors were able to redefine the superconducting dome in the phase diagram of Bi2212, present the first ARPES characterization of the very overdoped (OD) region of the superconducting dome and gain new insights regarding the importance of the Fermi surface topology for superconductivity in cuprates.

The Authors address the need for a tool to determine the doping level of Bi2212 crystals (and potentially for all the cuprates), instead of relying on the commonly employed (and not necessarily valid) universal relation between T_c and doping level. I agree that this is an interesting result for the cuprate community. Therefore, I consider this work suitable for the publication in Nature Communications after addressing/clarifying these following points.

1. The Authors claim that the “minimal gap loci” curves map the Fermi surface effectively, facilitating application of the Luttinger theorem. In addition, the Authors remark that this method is valid only if particle-hole symmetry is verified. However, there are several experiments which suggest that the pseudogap is not particle-hole symmetric (e.g. Ref. [12], discussed in the following). How would the particle-hole asymmetry of the pseudogap influence the extracted doping level (relative error)? How might different origins of the pseudogap impact the applicability of the Luttinger theorem?

Along the same line, the assumption of particle-hole symmetry at the antinode is not well-founded unless the pseudogap is a precursor to the superconducting state (which does not seem to be the case, as pre-formed Cooper pairs are detected for temperatures well below T^* , c.f. Ref. [26] or Reber et al., Nat Phys 8, 606 (2012)). Therefore, the assumption of particle-hole symmetry may lead to the extraction of an erroneous gap amplitude at the antinode. In addition, at the antinode both the superconducting gap and pseudogap are likely superimposed: consequently, I think that the plot of $2\Delta_0$ as the function of the doping in Fig.3a and Fig.3b is not relevant at this stage. There is already confusion within the community regarding determination of the antinodal gap value and its origin. I advise the Authors to reword/change this part of the manuscript.

However, I find it very interesting that for highly OD samples, i.e. where the pseudogap is absent and the $2\Delta_0$ value can be associated directly with the superconducting gap, the value of $2\Delta_0/kBT_c$ is close to what is expected for a d-wave BCS.

The referee is absolutely correct in pointing out that the pseudogap does not have to be related to pairing and therefore does not have to be particle-hole symmetric. The question of pseudogap's origin is a long standing one that is beyond the scope of our paper - we cannot offer more insights into this phenomenon than our ARPES results allow us. We limit our discussion and conclusions only to what our current ARPES results can clearly support. The determination of mid-gap position would require the knowledge of the upper gap edge for each k_F - an impossible task for ARPES at low temperatures where the occupation of these states is essentially zero. That is why we compare the measured contours at high temperature where no gaps are detected with those at low T , where we see the gap, but use the minimum gap loci for contour's construction. The fact that these contours are identical even in the underdoped regime is pretty remarkable. It might not necessarily imply that the low-temperature (total) gap (pseudogap superimposed onto superconducting gap) is particle-hole symmetric everywhere on the FS, but it would require some improbable coincidences to occur to keep the shape and the volume of these contours constant.

The naïve reasoning would imply that for the hole doped cuprates, the particle-hole asymmetry would affect the apparent hole count in the following manner: if the gap is centered below (above) E_F , the ARPES deduced hole count would be overestimated (underestimated). While it is clear that the number of mobile carriers, contributing to transport, will be necessarily affected by the (pseudo)gap, it is not clear why the total charge (Luttinger) count would change with temperature. Also, asymmetric gaps are usually associated with the breaking of translational symmetry (density waves) that reconstruct the electronic states, including the Fermi surface. As stated in the manuscript, we do not see any evidence of that in our data.

On the referees second comment that the antinodal gap does not reflect only the pairing gap – we certainly agree and we thought that it was clear from the manuscript that we talk about that gap as the total spectral gap measured at the antinode. We have now made that point even clearer.

We also agree with the referee that the observation of the gap approaching the BCS - d -wave limit when the pseudogap is absent is significant. To the best of our knowledge, this seems to be limited only to the highly overdoped, previously inaccessible regime, $p > 0.25$.

2. The Authors claim, in just one sentence, that they “do not see any change in shape between the low- and high-temperature Fermi contours that were reported in Ref. [12]”. This is a very important point, essential to validate the methodology proposed in the manuscript and should be discussed in more depth. I recommend the Authors add additional experimental evidence supporting their claim (for instance EDCs at various momenta and temperatures as shown in Ref. [12], as well as the suggestion below in minor point i.).

We have now added the figure for the UD87 sample, comparing the constant energy contours and several spectra along the relevant momentum lines at low (gapped states) and high (gapless states) temperatures.

3. The Authors do not provide error bars for the extracted doping levels (p). Error bars and how they have been determined should be added to the main text. In addition, how is the minimum gap loci determined (via EDCs or MDCs)? The Authors used a simplified tight-binding model from Ref. [13] but they neglect the (weak) k_z dependence: how could k_z affect the fitting and the relative error of the extracted doping level? How was the tight-binding fitted to the experimental data (by eye, maximum of the EDCs, etc...)? How did they distinguish bonding and antibonding states in very UD samples? Was the fitting procedure robust?

This is already discussed in reference to the questions of the first referee. In relation to the resolvability of the bonding-antibonding doublet, we note that these states were clearly resolved when their momentum width is less than the momentum splitting. As we state in the manuscript, this was the case in all but the most underdoped sample (UD30), at least in some regions of the k -space, when the maps were taken at low temperature. However, even for the most underdoped sample, using the same bi-layer splitting seems reasonable, as the experimental width of the superimposed intensity originating from the unresolved doublet follows the same $\sim(\cos k_x - \cos k_y)^2$ dependence as the splitting. We are aware that our ability to resolve the splitting, particularly in substantially underdoped samples, is somewhat surprising – we think that our way of cleaving (no epoxy) and preparation (annealing in UHV) of underdoped samples results in high surface flatness and uniformity and sharper photoemission spectra than what the conventional preparation/cleaving methods would usually produce. An additional illustration of remarkable quality of our data in the underdoped regime can be seen in Fig. 1 (C,G) and Fig S2(C,D), where the tiny anticrossing between the original bands and the diffracted replicas (due to the structural super-modulation) can finally be identified after more than 25 years of ARPES research on Bi2212. We intend to discuss this observation in more detail in one of our future publications.

4. The authors can determine the T_c for OD samples annealed in ozone only through spectroscopic measurements. This is done by tracking the evolution of the low-energy one-electron removal spectral function as shown in Ref. [26]. However, Ref. [26] is not cited correctly. Kondo et al. (Ref. [26]) have shown that the superconducting gap is filled rather than closed due to the loss of coherence of the macroscopic superconducting condensate, for both UD and OD samples. Therefore, the quasiparticle peak position is not really moving. On the contrary, quasiparticle spectral weight moves from the quasiparticle peaks into the gap. This is clear looking at Fig. 2G. The authors should reword this section. Ultimately, I agree that the determination of T_c via the leading edge position or the

intensity at the Fermi level is sufficiently accurate for the purpose of this manuscript.

Again, the referee is correct – we actually derive T_c from the leading edge position (same procedure as in Kondo et al).

Additional (Minor) Points:

i. The Authors discuss in the main text and in Methods that the mapping for $T > T^*$ shows exactly the same Fermi contours observed for $T < T_c$. They show in Fig. S2 a comparison of two FS for $T = 20$ K and $T = 140$ K, for Bi2212 OP91. Given the fact that this observation is in stark contrast with what was reported in Ref. [12], I would recommend the Authors to show the same comparison for the other OD samples they measured, as well as UD85.

As already noted above, we have now added the figure for UD87 sample, comparing the gapped and gapless constant energy contours and spectra showing dispersion in the superconducting and normal states with no detectable change in the momentum between the “underlying” gapped and gapless Fermi surface.

ii. References [14, 15] are placed during the discussion of the Luttinger count. These references are more appropriate for discussing the tight-binding model or the observed diffracted replicas and shadow bands.

The references are now moved to appropriate places and some new are added.

iii. Dorion-Leyraud and co-workers (Nat. Comm. 8:2044) have shown that the pseudogap appears only for dopings $p \leq p_{\text{Lifshitz}}$. In the present case, experimental data suggest the pseudogap in Bi2212 disappears for OD60/OD80 (corroborated also by SI-STM studies, Ref. [9], dashed red line in Fig. 3A) while the Lifshitz transition occurs at the end of the superconducting dome. I feel that this point should be mentioned and discussed in more depth. The first ARPES study of the very OD part of the phase diagram and the observation of the critical point associated with the end of the superconducting dome are definitely interesting. Differences between the critical points associated with the end of the superconducting dome and the pseudogap should be discussed.

Dorion-Leyraud et al, discuss the high field transport measurements on single-layer Nd-LSCO cuprate and relate the pseudogap disappearance with the change in FS topology (switching from hole- to electron-like). This is similar to the effects seen in Raman on Bi2212 (Benhabib et al). On the other hand, another type of Lifshitz transition has been proposed based on transport measurements in high magnetic field

in YBCO (Badoux et al [18]), where the transition is attributed to the change from small pockets to a large FS.

Although the changes in FS topology are more natural to connect to macroscopic transitions observed in systems with single Cu-O plane and single FS, we think that even in these systems, the situation is not as simple as that study suggests. Multilayer cuprates, and in particular, Bi2212 are more complicated as their electronic structure involves more than one state and more than one Fermi surface that can never display Lifshitz transition simultaneously. Our study on Bi2212 shows that the antibonding FS changes from hole like to electron like at $p \sim 0.29$, where $T_c \rightarrow 0$. The bonding state's FS remains hole-like and in general shows much less change with doping. We have now expanded the discussion of the interesting coincidence between the Lifshitz transition observed here and the disappearance of superconductivity at the end of main text.

iv. Comin et al. [22] did not report any observable reconstruction of the Fermi surface but they suggested that the CDW order originates from the nesting vector connecting the end of the Fermi arcs.

They do not observe it, but they claim that the reconstruction takes place. Their simulations certainly suggest the reconstructed Fermi surface, as should always be the case in the state with broken translational symmetry. However, we note that the width of their measured FS is such that nothing could be clearly resolved one way or the other, as it is often the case in studies where broken translational symmetry is claimed.

v. The Authors discuss the presence of the pseudogap at the antinode and list several possible origins (CDW, SDW, PDW, YRZ model) even if direct evidence of Fermi surface reconstruction is not observed (possibly due to the vanishing of the spectral weight). Additional scenarios where the pseudogap is induced by strong-electronic correlations, i.e. the localization/delocalization of charges associated with the proximate Mott insulating phase, should be discussed briefly.

In general, models emanating from proximate Mott state would not result in a pseudogap with d -wave like symmetry. The one exception that we are aware of is a cluster extension of DMFT, where a type of k -space resolved Mott transition occurs, resulting in nodal-antinodal dichotomy and a d -wave like reduction of spectral weight. We have now included a short statement and a reference to the work by Ferrero et al in the revised MS.

vi. As discussed previously, the superconducting gap for both OD and UD samples does not close at T_c (Ref. [26], Reber et al., Nat Phys 8, 606 (2012) or Boschini et al. Nat. Mat. 17, 416 (2018)). The superconducting gap closes at a higher temperature T_p different than T^* ($T_c < T_p < T^*$).

We agree. As stated before, we derive T_c from the leading edge position (same procedure as in Kondo et al).

vii. In the caption of Fig. 2: spectral (G) and (H) are symmetrized relative... (missing (H)).

This is now corrected.

REVIEWERS' COMMENTS:

Reviewer #1 (Remarks to the Author):

All the points raised in the previous round of review have been satisfactorily addressed. I think the paper can be published now.

Reviewer #2 (Remarks to the Author):

The Authors addressed well my major concerns/questions and they provided new experimental evidences in the supplementary Figure S2. In addition, the extended discussion of the correlation between the Lifshitz transition and superconductivity is definitely beneficial for the manuscript.

The manuscript deserves the publication in Nature Communications. However, few minor points should be addressed.

i) I recommend the Authors to reword/remove the sentence about the reconstruction in Bi2201 observed by Comin. In fact, Comin et al. did not observe any reconstruction and they did not claim any reconstruction induced by the folding of the Fermi surface driven by CDW (the term reconstruction is not used in their paper). Note that the report of a very short-range charge order does not directly imply the observation of a reconstruction. Comin et al. used a simulated FS based on the YRZ model (see their supplementary materials) to show that the nesting vector connecting the hot-spots of the Fermi surface matches well with the Q-vector of the CDW extracted via resonant x-ray scattering. In other words, they used a modeled YRZ spectral function to simply simulate the Fermi surface for considerations on the nesting vectors. Therefore, the claim that Comin et al. "observed" a reconstruction is definitely not correct and misleading.

ii) The Authors claim (lines 117-121) that the quasiparticle (QP) peak at the antinode is shifting as a function of the doping (correct and it is definitely convincing) and this happens also in the temperature dependence of OD and UD samples (Fig. 2 G-H). As I stated in my previous report, several experimental evidences have reported that the QP peak position is not shifting (i.e. the gap is not closing) but the in-gap spectral weight increases for temperatures $T \leq T_c$. The increase of the in-gap spectral weight is associated with a broadening of the spectral features which could result in an "apparent" movement of the QP peak (also for OD sample!), see references [25-26]. I fully understand that this technical discussion should not be included in the manuscript and I appreciate indeed the statement in line 97. However, I feel that the claim about the movement of the QP peaks may be misleading and I would recommend the Authors to clarify/reword this point. For instance, instead of QP peak position movement, the Authors could describe Fig. 2G-H in terms of an increase of the in-gap spectral weight (not taking part in the closure/filling debate).

iii) The sentence in lines 126-128 is not really clear and readable.

iv) Figure 2J shows the movement of the "apparent" QP peak position but the y-label indicates the gap amplitude. This is confusing. In fact, as I wrote in point ii), the position of the apparent QP peak may be not related to the gap amplitude.

v) Figure S2 is partially missing information about the temperature, energy and doping in

panels C, D, E. Moreover, the line in D and E is centered at $kx = -0.7 \text{ \AA}^{-1}$ (negative and missing unit).

vi) The Authors should clarify somewhere (methods or figure caption) that the TB-fit was done "by-eye" (and the associated uncertainty), as they discussed well in their reply to Reviewer #1.

REVIEWERS' COMMENTS:

Reviewer #1 (Remarks to the Author):

All the points raised in the previous round of review have been satisfactorily addressed. I think the paper can be published now.

Reviewer #2 (Remarks to the Author):

The Authors addressed well my major concerns/questions and they provided new experimental evidences in the supplementary Figure S2. In addition, the extended discussion of the correlation between the Lifshitz transition and superconductivity is definitely beneficial for the manuscript.

The manuscript deserves the publication in Nature Communications. However, few minor points should be addressed.

i) I recommend the Authors to reword/remove the sentence about the reconstruction in Bi2201 observed by Comin. In fact, Comin et al. did not observe any reconstruction and they did not claim any reconstruction induced by the folding of the Fermi surface driven by CDW (the term reconstruction is not used in their paper). Note that the report of a very short-range charge order does not directly imply the observation of a reconstruction. Comin et al. used a simulated FS based on the YRZ model (see their supplementary materials) to show that the nesting vector connecting the hot-spots of the Fermi surface matches well with the Q-vector of the CDW extracted via resonant x-ray scattering. In other words, they used a modeled YRZ spectral function to simply simulate the Fermi surface for considerations on the nesting vectors. Therefore, the claim that Comin et al. “observed” a reconstruction is definitely not correct and misleading.

We have now changed the sentence in question and “Also, the reconstruction that is observed in the single-layer compound $(\text{Bi,Pb})_2(\text{Sr,La})_2\text{CuO}_{6+\delta}$ by Comin \textit{et al} \cite{Comin2014}, but not by Kondo \textit{et al} \cite{Kondo2004}, does not show up in any of our samples.” is now replaced by:

“Also, the effects of charge ordering observed in the single-layer compound $(\text{Bi,Pb})_2(\text{Sr,La})_2\text{CuO}_{6+\delta}$ by Comin \textit{et al} \cite{Comin2014}, but not by Kondo \textit{et al} \cite{Kondo2004}, do not show up in any of our samples.”

ii) The Authors claim (lines 117-121) that the quasiparticle (QP) peak at the antinode is shifting as a function of the doping (correct and it is definitely convincing) and this happens also in the temperature dependence of OD and UD

samples (Fig. 2 G-H). As I stated in my previous report, several experimental evidences have reported that the QP peak position is not shifting (i.e. the gap is not closing) but the in-gap spectral weight increases for temperatures $T \leq T_c$. The increase of the in-gap spectral weight is associated with a broadening of the spectral features which could result in an “apparent” movement of the QP peak (also for OD sample!), see references [25-26]. I fully understand that this technical discussion should not be included in the manuscript and I appreciate indeed the statement in line 97. However, I feel that the claim about the movement of the QP peaks may be misleading and I would recommend the Authors to clarify/reword this point. For instance, instead of QP peak position movement, the Authors could describe Fig. 2G-H in terms of an increase of the in-gap spectral weight (not taking part in the closure/filling debate).

We agree with the referee – indeed, Fig. 2(i) actually already shows intensities at the QP peak and at the Fermi level (in-gap spectral weight), along with the LEG (extracted without symmetrization). We have also changed wording describing Fig. 2j and used “apparent QP peak position” instead “QP peak position”. In any case, the gapless state is achieved when the two-peak structure (in symmetrized spectra) turns into a single peak.

iii) The sentence in lines 126-128 is not really clear and readable.

The confusing part of the sentence (now lines 130-132) is now removed.

iv) Figure 2J shows the movement of the “apparent” QP peak position but the y-label indicates the gap amplitude. This is confusing. In fact, as I wrote in point ii), the position of the apparent QP peak may be not related to the gap amplitude.

The y-label in the Fig. 2(j) graph is now changed to “apparent QP energy”.

v) Figure S2 is partially missing information about the temperature, energy and doping in panels C, D, E. Moreover, the line in D and E is centered at $k_x = -0.7 \text{ \AA}^{-1}$ (negative and missing unit).

The missing information from Fig. S2 (now Fig.4) is now added in the caption. The sign error is now corrected, but the units are correct – wave vector π/a is set to 1, as defined earlier in the text and figures.

vi) The Authors should clarify somewhere (methods or figure caption) that the TB-fit was done “by-eye” (and the associated uncertainty), as they discussed well in their reply to Reviewer #1.

This is now done in Methods section.